# Short-Term Safety and Psychosocial Impact of the BNT162b2 mRNA COVID-19 Vaccine in Cancer Patients—An Italian Single-Center Experience

**DOI:** 10.3390/biomedicines11010165

**Published:** 2023-01-09

**Authors:** Irene Persano, Massimiliano Cani, Benedetta Del Rio, Giorgia Ferrari, Edoardo Garbo, Elena Parlagreco, Chiara Pisano, Valeria Cetoretta, Marco Donatello Delcuratolo, Fabio Turco, Alessandro Audisio, Cristina Cecchi, Gianmarco Leone, Valerio Maria Napoli, Valentina Bertaglia, Valentina Bianco, Enrica Capelletto, Carmen D’Amiano, Massimo Di Maio, Martina Gianetta, Silvia Novello, Francesco Passiglia, Giorgio Vittorio Scagliotti, Paolo Bironzo

**Affiliations:** 1Department of Oncology, University of Turin, San Luigi Gonzaga University Hospital, 10043 Orbassano, Italy; 2Department of Oncology, University of Turin, Mauriziano Hospital, 10128 Torino, Italy

**Keywords:** COVID-19 pandemic, SARS-CoV-2 infection, prevention strategies, COVID-19 vaccination, cancer patients, patients reported outcomes, thoracic malignancies

## Abstract

Safety data regarding BNT162b2 in cancer patients (CPs) are scarce. Herein we report the side effects (SEs), the adverse events (AEs), and the patient-reported outcomes (PROs) following BNT162b2 administration in CPs treated at the San Luigi Gonzaga University Hospital. All CPs who agreed to participate in our vaccination campaign received BNT162b2 and were included in the descriptive analysis. An anonymous questionnaire investigating the occurrence of SEs/AEs and PROs was administered to the study population 21 days after the first dose. Pearson’s chi-squared test was used to estimate the risk of experiencing SEs/AEs according to selected variables. A total of 997 patients were included in the study: 62.0% had stage IV cancer, and 68.8% were receiving an active treatment, of whom 15.9% were receiving immunotherapy. SEs/AEs were recorded in 37.1% of cases after the first dose and in 48.5% of cases after the second dose. The most common SEs were muscle pain/local rash (27.9% and 28%, after the first and second dose, respectively). Patients older than 70 years showed lower risk of SEs/AEs, while women showed a higher risk. Before receiving the vaccine, 18.2% of patients felt fearful and/or insecure about the vaccination. After the first dose, 57.5% of patients changed their feelings positively. Our data support the short-term safety of BNT162b2 in CPs, regardless of disease stage and concurrent treatments. Overall, the vaccination showed a positive impact on quality of life.

## 1. Background

The outbreak of the Coronavirus Disease 2019 (COVID-19), caused by the recently discovered severe acute respiratory syndrome coronavirus 2 (SARS-CoV-2), first identified in China, has rapidly spread worldwide. On 11 March 2020, the World Health Organization (WHO) declared the COVID-19 pandemic. Italy has been one of the most affected countries, with around four million reported cases and more than 120,000 deaths as of April 2021 [1]. This global crisis led to an international effort for rapid isolation and genome sequencing of SARS-CoV-2. On 21 December 2020, the European Medicines Agency (EMA) authorized BNT162b2, the first mRNA COVID-19 vaccine, for use in humans. The next day, BNT162b2 was approved by the Italian Medicines Agency (i.e., “Agenzia Italiana del Farmaco”, AIFA), and on 27 December, the Italian government launched a national COVID-19 vaccination campaign. Cancer patients (CPs), due to the immunosuppression associated with both the disease and cytotoxic treatments, showed high mortality rates from COVID-19 and were included in the categories to prioritize for vaccination [2,3,4,5]. For the most vulnerable groups, international guidelines recommended mRNA-based SARS-CoV-2 vaccines, as they do not contain live viruses and do not pose an immediate safety risk in the case of immunodeficiency [6,7]. However, while these vaccines have been shown to be safe and effective in the general population, data in immunosuppressed patients are still scarce [8,9,10]. The development of COVID-19 vaccines and the subsequent vaccination campaign also had a significant impact on the media industry, leading to an animated public debate. On 31 March 2021, the Department of Oncology of San Luigi Gonzaga University Hospital started promoting a vaccination campaign with BNT162b2 specifically directed to CPs. This observational analysis aims to identify putative CP subgroups with higher risk of developing side effects (SEs) and adverse events (AEs) after the first and the second dose of the BNT162b2 in a cohort treated at our institution. Moreover, we investigated the patient-reported outcomes (PROs) in terms of quality of information provided before receiving the vaccine, influence of social media and specialist consultations on patients’ opinion, alongside the overall impact of the vaccination on psychological wellness and social activities.

## 2. Patients

All cancer patients on active/planned treatment or on follow-up after radical treatment (within 5 years of diagnosis) at our institution were invited to participate in the vaccination campaign with BNT162b2. Patients who tested positive for SARS-CoV-2 after 1 January 2021 or with poor performance status (Eastern Cooperative Oncology Group (ECOG) >2 or Karnofsky score <50%), were not offered the vaccine, according to national and local guidelines. Patients were asked about their history of allergic reactions via telephone call before the vaccination. Patients with previous allergic reactions to the active ingredient or to any of the excipients of BNT162b2 were ineligible for the vaccination, whereas patients with a history of severe allergic reactions (e.g., anaphylaxis) to any other substance were referred for an allergy consultation.

## 3. Study Design

All eligible patients who agreed to participate in the vaccination campaign received the standard recommended schedule of BNT162b2, consisting of two intramuscular injections (30 µg per injection), 21 days apart, between 31 March and 10 May 2021. Transient reactogenicity events were reported as SEs, while any untoward medical occurrence related to the vaccine was reported as an AE. An ad hoc clinician-generated anonymous questionnaire investigating the occurrence of SEs, AEs, and the psychosocial impact of the vaccination was administered to the patients receiving the second dose of the vaccine during the observation period in the hospital (Appendix A). All patients who received two doses of BNT162b2 vaccine and agreed to complete the questionnaire were included in the descriptive analysis. The short-term SEs/AEs related to the second dose were investigated via a telephone questionnaire administered 4–7 days after. Along with the informed consent given for the vaccination, by compiling the anonymous questionnaire, the patients gave their informed consent for the observational prospective study.

## 4. Statistical Analysis

Patients’ characteristics were analyzed by descriptive statistics. Categorical data were summarized as frequency and percentage. Pearson’s chi-squared test was used to estimate the risk of SEs/AEs according to selected variables (age, sex, previous allergic reactions, previous SARS-CoV-2 infection, site of the primary tumor, active antineoplastic treatment, immunotherapy, SEs/AEs after the first dose). In order to assess any differences in the risk of developing SEs/AEs based on the primary tumor site, we stratified patients according to their cancer type and compared the resulting subgroups with the ones affected by thoracic malignancies, which was the most represented subgroup in our cohort. Multivariate analysis was only performed on variables that were significant on the univariate analysis. A *p*-value of 0.05 or less was considered statistically significant. Analyses were performed with IBM SPSS Statistics for Windows, Version 27.0, Armonk, NY, USA. Graphs were plotted using Microsoft Excel.

## 5. Results

### 5.1. Patients’ Characteristics

A total of 1610 patients were considered eligible for the vaccination. Among them, 103 (6.4%) refused the vaccination and 403 (25%) had already been administered at least one dose at the time of recruitment. Therefore, the first dose of BTN162b2 was administered to 1104 cancer patients. Due to worsened performance status (*n* = 19), diagnosis of SARS-CoV-2 infection (*n* = 2), SEs related to oncological treatments (*n* = 29), fever (*n* = 11), hospitalization (*n* = 9), cancer-related death (*n* = 4), or unknown reasons (*n* = 10), 84 patients who had received the first dose of the vaccine did not receive the second one and thus did not complete the vaccination schedule. Among the 1020 patients who received both doses, 997 agreed to complete the anonymous questionnaire and were included in our study (Figure 1). Patients’ characteristics are summarized in Table 1. Median age of the patients was 67 years (range 31–91), and 664 (66.6%) of them were males. Thoracic malignancies (lung or pleural) were the most common type of primary tumor (383 [38.4%]). Other primary tumors included: prostate (256 [25.7%]), gastrointestinal (119 [11.9%]), breast (71 [7.1%]), genitourinary (67 [6.7%]), endocrine (61 [6.2%]), gynecological (19 [1.9%]), and others (e.g., head and neck cancer and thymic cancer) (21 [2.1%]). A total of 618 patients (62%) had stage IV cancer, 277 (27.8%) had an early-stage disease, and 102 (10.2%) were disease-free. At the time of recruitment, 299 patients (30%) were in a 5-year follow-up program after being radically treated for cancer, 12 patients (1.2%) were newly diagnosed and were planned for a locoregional or systemic treatment, while 686 patients (68.8%) were on active antineoplastic treatment. Among the latter, 22 (3.2%) were receiving or received locoregional treatment (e.g., radiotherapy, ablative treatments, and surgery), while 664 (66.6%) were receiving a systemic treatment: hormone therapy (269 [39.2%]), targeted therapy (150 [21.9%]), chemotherapy (ChT) (114 [16.6%]), immune-checkpoint inhibitors (ICIs) (90 [13.1%]), combination of ChT-ICI (19 [2.8%]), or other systemic therapies (22 [3.2%]). Our study population included 49 patients who had recovered from SARS-CoV-2 infection, as shown by a negative result on a PCR-test before January 2021. The infection had been associated with no symptoms or mild symptoms in twenty-three (46.9%) and fourteen cases (28.6%), respectively, whereas twelve patients (24.5%) had been hospitalized for COVID-19-related pneumonia, of whom two patients had developed severe respiratory failure requiring admission to the Intensive Care Unit (ICU). Before receiving the first dose, 33 patients were referred to allergy consultation, which resulted in 14 of them being prescribed a prophylactic antihistamine therapy.

### 5.2. Risk of SEs/AEs after the First and the Second Dose

A total of 370 patients (37.1%) experienced SEs succeeding the first dose of BTN162b2 and 484 (48.5%) after the second dose. Data regarding SEs after the second dose have not been collected for 47 patients due to the patients not answering our phone calls. SEs after the first and second dose are summarized in Figure 2. Local SEs at the site of injection were the most reported, with 278 (27.9%) and 279 (28%) patients experiencing pain at the injection site and/or local rash after the first and the second dose, respectively. For both doses, the most common systemic SEs were fatigue (104 [10.4%]; 167 [16.7%], for first and second dose, respectively), arthralgia (89 [8.9%]; 121 [12.1%]), headache (46 [4.6%]; 60 [6%]), and fever (25 [2.5%]; 112 [11.2%]). Lymphadenopathy was reported by 0.5% of patients following the first dose and 0.6% after the second dose. Allergic reactions (rash and pruritus) were reported in few cases (9 [0.9%]; 8 [0.8%]), and no severe reactions (e.g., anaphylaxis) were observed. None of SEs/AEs required a special intervention or hospitalization. No vaccine-related deaths were reported. Patients older than 70 years had a lower risk of developing SEs/AEs after the first dose [OR 0.47 (95%CI 0.36–0.62), *p* < 0.0001], both according to univariate and multivariate analyses. Women are at increased risk of presenting SEs/AEs after the first dose compared to men according to univariate analysis [OR 2.48 (95%CI 1.89–3.25); *p* < 0.0001], and this was confirmed by the multivariate analysis, although the magnitude of difference was smaller [OR 1.23 (95%CI 1.14–1.32); *p* < 0.001]. Patients with previous SARS-CoV-2 infection showed a higher risk of developing SEs/AEs after the first vaccine dose both according to the univariate [OR 1.79 (95%CI 1.01–3.18); *p* = 0.048] and multivariate analyses, but with a smaller magnitude of difference (OR 1.16, 95%CI 1.02–1.33, *p* = 0.029). A positive history for allergic reactions emerged as a significant risk factor for developing SEs/AEs following the first dose when considered as an independent variable [OR 1.72 (95%CI 1.21–2.44); *p* = 0.002]. However, the difference was not significant according to the multivariate analysis (*p* = 0.065). Patients older than 70 years showed a lower risk of developing SEs/AEs after the second dose [OR 0.53 (IC 95% 0.41–0.69); *p* < 0.0001]. The multivariate analysis results confirmed this finding, although with smaller magnitude [OR 0.89 (95%CI 0.84–0.95); *p* < 0.001]. Women are at increased risk of presenting SEs/AEs after the second dose compared to men [OR 1.89 (95%CI 1.43–2.49); *p* < 0.0001]. When adjusted for all the considered variables, this difference was statistically significant, although smaller [OR 1.14 (95%CI 1.05–1.23); *p* < 0.001]. According to the univariate analysis, a positive history for allergic reactions was associated with a higher risk of developing SEs/AEs after the second dose [OR 1.44 (IC 95% 1.01–2.07); *p* = 0.047]. However, this result was not confirmed by the multivariate analysis (*p* = 0.37). Patients who experienced SEs/AEs after the first dose showed an increased risk of developing SEs/AEs after the second dose [OR 3.10 (95%CI 2.35–4.10); *p* < 0.0001]. This difference was statistically significant according to the multivariate analysis [OR 1.26 (95%CI 1.18–1.35) *p* < 0.001] (Table 2 and Table 3).

### 5.3. Psychosocial Impact of the Vaccination on Cancer Patients

Most of the patients included in the study were hopeful (708 [71.8%]) and thrilled (249 [25.3%]) in reference to the vaccination prior to the first dose administration, while 124 patients (12.6%) expressed fear, 70 (7.1%) expressed insecurity, 110 (11.2%) felt indifference, and 11 (1.1%) patients did not answer the question. After completing the vaccination, only 51 (5.2%) and 36 (3.6%) patients felt fearful and insecure, respectively, while the majority (847 [85.8%]) expressed hope and/or enthusiasm. A total of 793 (79.3%) patients declared an improvement in terms of confidence when carrying out social activities in at least three out of four of the considered domains (visiting public places, spending time with family/friends, attending check-up visits in public health services, and practicing recreational and sport activities). For 104 (10.5%) patients only, the vaccination had no significant impact in their social activities/quality of life. The opinions regarding the vaccination in the population included in our study were influenced by the consultation with general practitioner/oncologist in 386 cases (47.5%), by mass-media in 257 cases (31.7%), by family/friends in 103 cases (12.7%), and by the scientific literature in 65 cases (8%), although 185 (18.6%) patients did not complete this part of the questionnaire. Information provided about the vaccination prior to recruitment in our campaign was considered adequate by 855 (85.8%) patients, confused by 80 (8.0%) patients, and insufficient by 39 (3.9%) patients.

## 6. Discussion

CPs are a frail population with a high risk of exposure to SARS-CoV-2 virus, due to the regular access to the hospitals for medical care, and high mortality rate from COVID-19. An Italian study showed that 20% of patients who died from COVID-19 in Italy had active cancer [11]. Similarly, another analysis confirmed these findings, showing a death rate of 13% in CP vs. 1.4% in a Chinese unselected population [2,12]. Furthermore, the pandemic overwhelmed healthcare systems worldwide and, in some cases, led to delay in primary diagnosis and therapeutic access for CPs, with a likely impact on cancer morbidity and mortality [13]. Hence, preventive strategies, such as an efficient vaccination campaign, are crucial to ensure protection and continuity of care for this high-risk group. Oncology societies promptly faced the issue by developing specific guidelines for cancer care during the pandemic. Notably, the National Comprehensive Cancer Network (NCCN), COVID-19 Vaccination Advisory Committee, the European Society for Medical Oncology (ESMO), and Associazione Italiana Oncologia Medica (AIOM) representatives recommended that patients with active cancer and those on antineoplastic treatment should be prioritized for the vaccination [6,7,14]. Upon launching the national vaccination campaign, given the initial limited supply of COVID-19 vaccines, the National Strategic Plan for Vaccination (NSPV) defined high-priority categories: healthcare workers, staff and hosts of nursing homes, people of age 80 and over, and vulnerable patients due to organ damage and/or immunosuppression, including those affected by solid and hematological tumors [5,15]. In order to facilitate and accelerate our patients’ access to COVID-19 vaccines, the Department of Oncology of San Luigi Gonzaga University Hospital started a vaccination campaign dedicated to CPs on 31 March 2021. Since at the time of recruitment recommendations suggested mRNA-based SARS-CoV-2 vaccines for frail groups, CPs enrolled in our study were vaccinated with BNT162b2 [6,7]. Overall, mRNA-based vaccines have shown an efficacy of more than 90% in preventing COVID-19 disease, with an optimal safety profile in the general population [6,7]. Despite this reassuring data, a major concern regarding the immunodeficiency of CPs, related to both the disease itself and oncological treatments, was raised among the cancer scientific community. Indeed, immunodeficiency could lead to suboptimal efficacy of the vaccination in CPs [16]. However, three prospective, longitudinal, observational studies on immunogenicity of BNT162b2 in CP proved effective antibody response, especially in solid tumors, within two weeks after the early (day 21) second vaccine dose [10,17,18]. In our vaccination campaign, more than 92% of patients who received the first dose completed the vaccination with the second boost after 21 days. This was reassuring data considering that most of these patients did not receive the second dose of vaccination due to cancer-related events (e.g., performance status worsening, SEs due to cancer treatment). Nonetheless, the percentage of patients who achieved a protective immune response remains unknown, since seroconversion to SARS-CoV-2 spike (S) protein was not assessed in our study, and outcome data (e.g., COVID-19 disease, hospitalization, death) among fully vaccinated cancer patients are not available yet. Even though recent evidence pointed out that a single dose of mRNA vaccine elicited rapid immune responses in seropositive people, the Italian guidelines recommend the full vaccination schedule for immunocompromised patients with prior documented SARS-CoV-2 infection [19]. Therefore, seropositive CPs included in our study received both doses, regardless of the severity of previous SARS-CoV-2 infection. Safety data regarding BNT162b2 administration in immunocompromised patients are fragmented and based on case reports and two single-center prospective series [10,15].

The purpose of this study was to describe the incidence and magnitude of SEs/AEs after administration of an mRNA-based COVID-19 vaccine in a prospective large cohort of patients affected by solid malignancies. In line with data from BNT162b2 pivotal trials, SEs were more frequently reported after the second dose, and the most common SE was pain at the injection site. Although the nature of SEs reported by our patients is similar to what was already described in the aforementioned trials, the prevalence of SEs is lower in our study. Remarkably, fatigue was experienced by 16.7% of CPs after the second dose versus 51% of healthy individuals aged 55 and older [8]. This marked difference in prevalence may be due to supportive therapies (e.g., corticosteroids and analgesic therapy) often given to CPs, which could have masked the SEs related to the vaccine. Another possible explanation is that the lower prevalence of SEs in CPs is associated with a weaker immune response in this immunodeficient population. Lastly, a selection bias cannot be excluded, considering the overlap between side effects from the vaccine and common cancer-related symptoms (e.g., fatigue). AEs related to the vaccine (lymphadenopathy and mild allergic reactions) were rare (<1%), and we did not observe any case of anaphylaxis, despite it being previously reported [20]. Younger patients seem to have a higher reactogenicity to the vaccine, showing an increased incidence of SEs/AEs after both doses. This finding is consistent with what was previously reported in pivotal clinical trials, although comparison between those trials and our study is limited by the selection of a different cut-off to split the two populations (70 years and 55 years old for our study and pivotal trials, respectively). Safety analyses conducted by the Centers for Disease Control and Prevention (CDC) after the first month of vaccination with mRNA vaccines in the United States revealed that 78.7% of AE reports submitted were observed in women. Accordingly, our analysis showed a significant increased risk for developing SEs/AEs in women compared to men. Sex differences in the response to vaccination, in terms of magnitude of immune responses and severe AEs following immunization, have already been established [21]. However, further studies are needed to confirm these data in anti-SARS-CoV-2 mRNA-based vaccines. SEs/AEs were reported more often by seropositive CPs than seronegative ones after the first dose, whereas there is no difference between the two groups following the second dose. Although data on the safety of anti-SARS-CoV-2 vaccines in CPs with previous SARS-CoV-2 infection are limited, our findings appear to be in contrast with what has been already described in the general population. Indeed, a prospective study of reactogenicity, safety, and antibody response after one and two doses of mRNA vaccine in seronegative and seropositive healthcare workers showed that the second injection generates a greater overall systemic reaction than that observed after the first one, regardless of the initial serological status of the participants [22]. A potential bias that could explain our result is that seropositive patients who developed severe SEs/AEs after the first dose were considered ineligible for the second dose. However, the proportion of seropositive CPs who received the first and second dose does not differ significantly (4.9% vs. 4.4%, respectively). Further data are needed to assess the efficacy and safety in this specific cohort. Another common concern was related to potential interactions between the vaccine and antineoplastic treatments. In particular, some concerns have been raised about the potential enhancement of vaccine immune-related SEs in patients treated with ICIs. A previous report on 170 patients treated in an Israeli institution supports the short-term safety of BNT162b2 in CPs on treatment with ICIs. In this study, the investigators did not observe either new immune-related AEs or exacerbation of pre-existing immune-related AEs [23]. These findings are consistent with what we observed in our cohort of CPs. Specifically, patients who were receiving an active treatment, included immunotherapy, at the time of the vaccination did not show a higher risk of developing SEs/AEs following the two doses of vaccine.

Despite the fact that the positive effect of vaccination campaigns on the COVID-19 pandemic worldwide is indisputable, the opportunity of imposing a vaccination on the entire adult population opened a public debate on the risks/benefits balance. Indeed, according to recent data published by “Fondazione Italia in Salute”, 7.5% of Italian citizens are strongly against the vaccination, while 9.9% are hesitant [24]. A primary objective of our analysis was to describe how a special subgroup of patients, such as CPs, was affected by the vaccination. The proportion of patients who refused the vaccination (6.4%) in our study is slightly lower compared to the compliance in the general population. Furthermore, our data showed that only a small number of patients who joined the vaccination campaign were afraid of vaccine-related SEs before receiving the first dose, while the majority expressed a “positive vaccine sentiment”, with a significant percentage of patients who positively changed their feelings after the second dose. Patients’ opinion on the vaccination was mainly influenced by the general practitioner/specialist, and the information provided before recruitment was in most cases adequate and complete. Indeed, a frail population with high-intensity medical care need, such as CPs, mostly relies on healthcare providers, who should inform the patients about the increased risk of complications from COVID-19, reassure them about the risks/benefits balance, and ultimately encourage the vaccination. The vaccination showed a deep positive impact in patients’ quality of life, meaning that fully vaccinated patients feel more comfortable and confident about their daily activities. Nonetheless, the proportion of CPs who refuse the vaccine is still significant, and further efforts by the governments, scientific societies, and clinicians should be made to raise awareness among the population.

The main limitation of our study is that it was conducted in a single center and only enrolled patients with solid malignancies, most of them affected by thoracic tumors. Therefore, our findings may not be applicable to other centers with a different distribution in cancer types and variable rates of COVID-19 incidence. Moreover, due to the short follow-up, our observational analysis did not assess relevant endpoints, such as overall survival and long-term safety. Another limitation of our study is the absence of data about psychiatric comorbidities and their specific treatment, as well as the educational level of enrolled patients. These data may have contributed to the psychosocial impact evaluation. Nonetheless, our study gathered reassuring safety data on BNT162b2 in CPs regardless of the active treatment, being the largest dedicated prospective trial to our knowledge at the at the time of submission. Further large-scale, multicentric studies with longer follow-up are required to confirm our results and assess the long-term safety of mRNA COVID-19 vaccines in this specific population.

## 7. Conclusions

Our analysis supports the optimal short-term safety of BNT162b2 in a large prospective cohort of patients with solid tumors, regardless of the active treatment, including patients receiving immunotherapy. CPs mostly relied on the healthcare providers’ recommendations and have a “positive vaccine sentiment”. Moreover, the vaccination showed a positive psychological and social impact in this frail population.

## Figures and Tables

**Figure 1 biomedicines-11-00165-f001:**
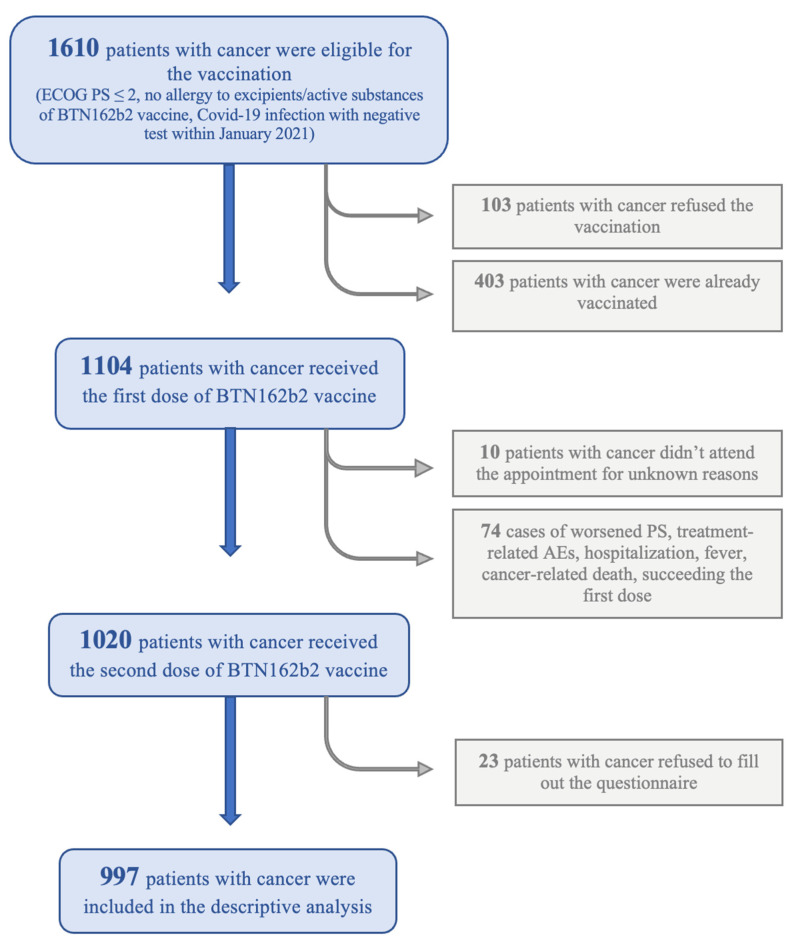
Flow chart of the recruiting in the vaccination campaign with BTN162b2 vaccine carried out at the Department of Oncology of San Luigi Gonzaga University Hospital and the enrollment in the descriptive analysis. PCR: Polymerase Chain Reaction; ECOG PS: Eastern Cooperative Oncology Group Performance Status; AE: Adverse Events.

**Figure 2 biomedicines-11-00165-f002:**
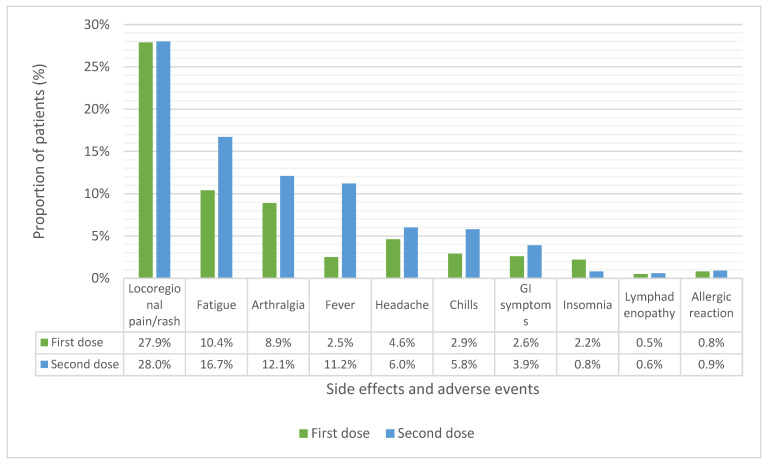
Local and systemic side effects and adverse events after the first and the second dose of BTN162b2 vaccine in cancer patients treated at our institution. GI: gastrointestinal.

**Table 1 biomedicines-11-00165-t001:** Main characteristics of the study population. HT: hormone therapy; TT: targeted therapy; ChT: chemotherapy; ICIs: immune checkpoint inhibitors.

Characteristics	Number of Patients (%)
Age	
<70 y	501 (50.3)
≥70 y	496 (49.7)
Gender	
Males	664 (66.6)
Females	333 (33.4)
Type of malignancy	
Thoracic	383 (38.4)
Gastrointestinal	119 (11.9)
Prostatic	256 (25.7)
Breast Cancer	71 (7.1)
Genitourinary	67 (6.7)
Endocrine tumor	61 (6.2)
Gynecological	19 (1.9)
Others	21 (2.1)
History of SARS-CoV-2 infection	49 (4.9)
Asymptomatic	23 (46.9)
Mild symptoms	14 (28.6)
Mild/moderate SARS-CoV-2 pneumonia	10 (20.4)
Severe pneumonia	2 (4)
Allergy history	150 (15)
Allergy to excipients or active substances of BNT162b2 vaccine	0 (0)
History of severe allergic reactions	33 (22)
Prophylaxis with antihistamine therapy	14 (9.3)
Disease status	
Early-stage disease	618 (62)
Advanced disease	277 (27.8)
Disease-free	102 (10.2)
Active treatment	686 (68.8)
Locoregional treatment	22 (3.2)
HT	269 (39.2)
TT	150 (21.9)
ChT	114 (16.6)
ICIs	90 (13.1)
ChT-ICIs	19 (2.8)
Other	22 (3.2)
5-year follow-up	299 (30)
Newly diagnosed	12 (1.2)

**Table 2 biomedicines-11-00165-t002:** Proportion of side effects and adverse events according to all the selected variables after the first vaccine dose (A) and after the second vaccine dose (B).

**A**		
	**No. of Patients with Side Effects** **or Adverse Events**	**Proportion** **(95% CI)**
All patients	370/997	37.1% (34.2–40.2%)
Age < 70 years	229/501	45.7% (41.5–50.2%)
Age ≥ 70 years	143/496	28.8% (25.0–33.0%)
Men	200/664	30.1% (26.8–33.7%)
Women	172/333	51.7% (46.3–57.0%)
Thoracic tumors	137/383	35.9% (31.2–40.8%)
GI tumor	46/119	39.0% (30.7–48%)
Breast cancer	40/71	55.7% (44.1–66.8%)
Prostate cancer	82/256	32.2% (26.7–38.1%)
Urothelial/renal cancer	25/67	37.3% (26.7–49.3%)
Endocrine system cancer	28/61	45.2% (33.4–57.5%)
Gynecological cancer	11/19	57.9% (36.3–76.9%)
Other sites	4/21	20.0% (8.1–42%)
Active antineoplastic treatment	254/686	37.1% (33.5–40.8%)
Immunotherapy	32/90	35.8% (27.4–45.1%)
Previous SARS-CoV-2 infection	25/49	51.0% (37.5–64.4%)
Previous allergic reactions	73/150	48.7% (40.8–56.6%)
**B**		
	**No. of Patients with Side Effects** **or Adverse Events**	**Proportion** **(95% CI)**
All patients	484/997	48.6% (45.5–51.7%)
Age < 70 years	280/501	55.8% (51.4–60.1%)
Age ≥ 70 years	214/496	43.2% (38.8–47.7%)
Men	292/664	44.0% (40.3–47.8%)
Women	205/333	61.5% (56.0–66.8%)
Thoracic tumors	201/383	52.4% (47.4–57.5%)
GI tumor	63/119	53.1% (44.0–62.1%)
Breast cancer	36/71	50.7% (38.4–61.6%)
Prostate cancer	120/256	46.7% (40.5–53.0%)
Urothelial/renal cancer	37/67	54.7% (42.3–66.3%)
Endocrine system cancer	30/61	49.1% (36.6–61.7%)
Gynecological cancer	13/19	70.6% (46.9–86.7%)
Other sites	9/21	45.0% (25.8–65.8%)
Active antineoplastic treatment	359/686	52.3% (48.5–56.2%)
Immunotherapy	51/90	57.1% (47.6–66.2%)
Previous SARS-CoV-2 infection	28/49	56.8% (42.2–70.3%)
Previous allergic reactions	88/150	58.7% (50.6–66.5%)
SEs/AEs to the first dose	252/370	68.1% (63.0–72.8%)

**Table 3 biomedicines-11-00165-t003:** Univariate and multivariate analyses and odds of SEs/AEs after the first dose (A) and after the second dose (B) based on all selected variables. GI: gastrointestinal; CP: cancer patients; SEs: side effects; AEs: adverse events.

**A**				
**Variables**	**Univariate Analysis**		**Multivariate Analysis**	
	**Odds Ratio** **(95% CI)**	***p*-Value**	**Odds Ratio** **(95% CI)**	***p*-Value**
Age ≥ 70 y vs. < 70 y	0.47 (0.36–0.62)	<0.0001	0.86 (0.81–0.91)	<0.001
Females vs. males	2.48 (1.89–3.25)	<0.0001	1.23 (1.14–1.32)	<0.001
Primary tumor site (vs. thoracic malignancies)				
GI tumor	1.14 (0.75–1.75)	0.54	1.05 (0.96–1.16)	0.29
Breast cancer	2.25 (1.34–3.77)	0.002	1.10 (0.97–1.25)	0.12
Prostate cancer	0.85 (0.61–1.19)	0.33	1.09 (1.00–1.19)	0.06
Urothelial/renal cancer	1.06 (0.62–1.82)	0.82	1.03 (0.92–1.17)	0.60
Endocrine system cancer	1.47 (0.86–2.53)	0.16	1.05 (0.93–1.19)	0.44
Gynecological cancer	2.46 (0.97–6.26)	0.06	1.17 (0.94–1.45)	0.15
Other sites	0.45 (0.15–1.36)	0.16	0.90 (0.73–1.12)	0.34
Antineoplastic treatment vs. no active treatment	0.94 (0.71–1.24)	0.66	0.98 (0.91–1.05)	0.48
Immunotherapy vs. other treatments	0.92 (0.61–1.40)	0.70	1.05 (0.94–1.16)	0.39
Previous SARS-CoV-2 infection vs. negative history of SARS-CoV-2 infection	1.79 (1.01–3.18)	0.048	1.16 (1.02–1.33)	0.029
Previous allergic reactions vs. non-allergic CP	1.72 (1.21–2.44)	0.002	1.08 (0.99–1.17)	0.065
**B**				
**Variables**	**Univariate Analysis**		**Multivariate Analysis**	
	**Odds Ratio** **(95% CI)**	***p*-Value**	**Odds Ratio** **(95% CI)**	***p*-Value**
Age ≥ 70 y vs. < 70 y	0.53 (0.41–0.69)	<0.0001	0.89 (0.84–0.95)	<0.001
Females vs. males	1.89 (1.43–2.49)	<0.0001	1.14 (1.05–1.23)	0.001
Primary tumor site (vs. thoracic malignancies)				
GI tumor	1.03 (0.67–1.57)	0.90	1.02 (0.92–1.13)	0.66
Breast cancer	0.91 (0.54–1.52)	0.71	0.86 (0.76–0.98)	0.03
Prostate cancer	0.79 (0.57–1.10)	0.16	1.01 (0.92–1.11)	0.85
Urothelial/renal cancer	1.09 (0.64–1.86)	0.74	1.03 (0.91–1.17)	0.62
Endocrine system cancer	0.88 (0.50–1.53)	0.64	0.94 (0.82–1.07)	0.35
Gynecological cancer	2.18 (0.75–6.30)	0.15	1.10 (0.87–1.39)	0.41
Other sites	0.74 (0.30–1.83)	0.52	1.02 (0.82–1.27)	0.86
Antineoplastic treatment vs. no active treatment	1.17 (0.89–1.54)	0.25	1,04 (0.97–1.12)	0.25
Immunotherapy vs. other treatments	1.32 (0.88–1.99)	0.18	1.05 (0.94–1.17)	0.41
Previous SARS-CoV-2 infection vs. negative history of SARS-CoV-2 infection	1.27 (0.69–2.34)	0.44	1.02 (0.88–1.18)	0.81
Previous allergic reactions vs. non-allergic CP	1.44 (1.01–2.07)	0.047	1.04 (0.96–1.13)	0.37
SEs/AEs after the first dose vs. no SEs/AEs after the first dose	3.10 (2.35–4.10)	<0.0001	1.26 (1.18–1.35)	<0.001

## Data Availability

Data is contained within the article or Appendix A.

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
