# Peer review of "Short-Term Safety and Psychosocial Impact of the BNT162b2 mRNA COVID-19 Vaccine in Cancer Patients—An Italian Single-Center Experience"

_biomedicines, 2023, doi:10.3390/biomedicines11010165_

Round 1
Reviewer 1 Report
In this paper, authors analyzed side effects (SEs), adverse events (AEs), and the cancer patient-reported outcomes (PROs) following BNT162b2 administration. This study provides real-life data on cancer patients and can provide more efficiency and accurate information about COVID-19. I believe that the study will contribute to the literature.
Some remarks:
1. The authors should report the presence of SARS-CoV-2-binding antibodies in all cancer patients at baseline and on day 21.
Line 66-67
2. “All cancer patients on active/planned treatment or on follow-up after radical treatment (within 5 years from diagnosis) at our Institution were invited to participate in the vaccination campaign with BNT162b2”
Have the type of treatment influenced the adverse effects reported by the patients?
3. In my opinion, other than cancer, age-related physiological alteration can influence the l response of the patients to treatment. Why the authors stratified the sample only >70; <70?
Author Response
1) We thank Reviewer#1 for this valuable comment on our study. Unfortunately, as we stated in line 253, this study was not designed to assess seroconversion to SARS-CoV-2 spike protein. Therefore, we cannot provide any data about this topic.
2) We thank Reviewer#1 for this insightful consideration. We included "active treatment" and "immunotherapy" as selected variables in the univariate and multivariate analyses of SEs/AEs. As reported in Table 2 and line 303-311, we found that patients who were receiving an active treatment, included immunotherapy, at the time of the vaccination did not show a higher risk of developing SEs/AEs.
3) We thank Reviewer#1 for this interesting comment. We stratified the patients in two categorical variables in order to apply the same chi-squared test to all considered variables. Although BNT162b2 pivotal trial used 55 y as cut-off, we chose a higher cut-off according to historical clinical studies conducted in elderly patients with cancer (see for example Gridelli C. Oncologist. 2001; Battaglin F, et al. BMC Cancer. 2018)
Reviewer 2 Report
very important manuscript. He discusses the issues of vaccination against COVID-19 in the population of cancer patients. The authors assess the issues of disagreement as well as the course of the post-vaccination period itself, taking into account side effects.
The work is very valuable, but I think it would be good if it was a bit extended.
- describing the pain at the puncture site - what was its intensity - are data available?
- whether there were patients with depression in the subgroups. Were the patients evaluated by a psychologist and/or psychiatrist to confirm the diagnosis.
- whether drugs affecting the depressive mood were used, from popular groups, e.g. b-adrenergic receptor antagonists, but not only these.
- what was the level of education of the patients?
Author Response
1) We thank Reviewer#2 for his comment. As reported in the questionnaire administered to the patients (suppl. mat.), we investigated the presence or absence of SEs/AEs, but we did not assess the intensity of each item in order to make the questionnaire more feasible in our real-life population. We considered local reactogenity as described in the BNT162b2 pivotal trial (Polack et al, NEJM 2020): mild-to-moderate pain at the injection site/redness/swelling.
2-3) We thank Reviewer#2 for this valuable comment. Unfortunately, psychiatric/psychological disorders were not assessed by specialist physicians in our real-life population. The questionnaire was given directly to the patients without any intermediation by clinicians, so this kind of assessment, despite of high interest, would probably not have been attainable and reliable. Our analysis just focused on psycho-social impact of the vaccination. We acknowledge that psychiatric disorders (as well as the specific treatment) may have affected somehow our results. We add this in the discussion section, underlining the lack of these data as a limitation of our study.
4) We thank Reviewer#2 for this wit comment. Patients were not asked about their educational level in the questionnaire. This is another possible limitation that we add in the discussion section of our manuscript.
Reviewer 3 Report
The paper titled "Short-term safety and psychosocial impact of the BNT162b2 mRNA COVID-19 vaccine in cancer patients. An Italian single-centre experience" deals with an interesting topic. The purpose of this study was to identify putative cancer patients subgroups with higher risk of developing side effects and adverse events after the first and the second dose of the BNT162b2 in a cohort treated at their Institution. Moreover, they investigated the patient reported outcomes in terms of quality of information provided before receiving the vaccine, influence of social media and specialist consultations on patients’ opinion, alongside the overall impact of the vaccination on psychological wellness and social activities
The paper is well-written but it needs some changes and clarifications:
Abstract - line 23: the percentage of patients that were receiving an active treatment is reported as 68.1% but in Table 1 it is 69%. Please, resolve this inconsistency.
Study design – lines 80-81: please, better explain the sentence "All vaccinated patients were included in the descriptive analysis." since the analysis considered 997 patients (who received two doses of vaccine and also agreed to fulfil the questionnaire) and not 1020 patients who had only received two doses of vaccine.
Results – Patient’s characteristics:
Line 118: the percentage of the most common type of primary tumour is reported as 34.8% but in Table 1 it is 38.4%. Please, resolve this inconsistency.
Line 120: the number of the endocrine tumour is reported as 61 but in Table 1 it is 62. Please, resolve this inconsistency.
Line 121: the number and the percentage of others tumours are reported as 21 [2.1%] but in Table 1 they are reported as 20 (2.0). Please, resolve this inconsistency.
Line 125: the percentage of patients in active antineoplastic treatment is reported as 68.1% but in Table 1 it is 69%. Please, resolve this inconsistency.
Table 1: please, add the percentage values next to the numbers of the characteristics “5-years follow-up” and “Newly diagnosed”.
Figure 1: patients with cancer included in the descriptive analysis are 997 and not 977 as it is reported in the last box.
Results – Risk of SEs/AEs after the first and the second dose:
Lines 172-184: the results relating to the univariate and multivariate analysis after the second dose and odds of SEs/AEs are not reported in Table 2, as is indicated in line 184. Table 2 B would seem to be missing. Please, resolve this inconsistency.
Figure 2: please, write decimal numbers using the point instead of the comma.
Moreover, the references are written in most of the paper incorrectly because they are superscript and reported in Roman numerals and not in Arabic numerals as instead they are listed at the end of the manuscript. According to the instructions for the authors “reference numbers should be placed in square brackets [ ], and placed before the punctuation." Please, write better references.
Author Response
1) We thank Reviewer#3 for his comment. We recalculated and corrected the percentages in the manuscript.
2) We thank Reviewer#3 for his observation. In the “Results – patients’ characteristics” section and Figure 1 we stated that only the patients who completed the anonymous questionnaire were included in our study. We will better specify the inclusion criteria also in “Study design section”.
3) We thank Reviewer#3 for these observations. Most of the inconsistencies reported in the “Results” section were typos. We apologize and recalculate all the percentages in Table 1 and in the text.
4) We thank Reviewer#3 for his comment. Of course both tables are available but there might have been an error in the file uploading. We apologize and uploaded Table 2 part B in the revised manuscript.
5) We thank Reviewer#3 for this comment. We wrote all the references in Arabic numerals, but probably the format changed after the uploading. We apologize if the numbers were not placed in square brackets, as indicated in the instructions. We corrected as required.
Reviewer 4 Report
This manuscript describes the short-term side effects and adverse events of cancer patients in Italy who received two-dose series of covid-19 vaccine. Given that established guidelines on covid-19 vaccines already exist and have been implemented, the novelty and importance of the information presented in this manuscript is unclear. There are multiple parts in the manuscript where information seems to be missing or misleading as follows:
Lines 18-20: This sentence appears incorrect. Figure 1 shows that 1104 patients received the first dose and 1020 patients received the second dose. Table 1 include 997 patients. These numbers do not match and please clarify.
Figure 1: Please clarify the total number of 977 in the last box given that Table 1 shows descriptive data for 997 cancer patients.
Table 2: Please clarify or add data after first dose (A) and after the second dose (B) as indicated in the title of the table. Please also clarify whether these odds ratios are based on univariate or multivariate analysis and please add odds ratios from both analyses. In text in the results section, please interpret the magnitude of the association based on multivariate analyses.
Figure 2: Using forest plots to present this type of data is confusing. Please use other format. Table works better.
Lines 291-296: It seems that data presented in Figure 2 shows the consistent results from this prospective study. Please clarify.
Lines 330-341: Figure 1 shows multiple deaths occurred after the first dose of vaccine. Please include more about these patients who have deceased and incorporate this information in the Results and Discussion sections.
Author Response
1) We thank Reviewer#4 for this comment. Among the 1020 patients who receives both doses, only 997 fulfilled the questionnaire. Characteristics of these patients are reported in Table 1.
2) We thank Reviewer#4 for his comment. That was actually a typo, so the total number of patients is 997. We apologize and correct this error.
3) We thank the Reviewer#4 for this comment. We have modified the table according to the Reviewer suggestion. In the results section, we have modified the text adding the results of the multivariate analyses in order to show the magnitude of the association.
4) We have modified the presentation of data according to Reviewer’s suggestion. Now the proportion of adverse events in the subgroups is presented in a table instead of the forest plot.
5) We apologize with the Reviewer#4 but we did not fully understand this point: “Lines 291-296: It seems that data presented in Figure 2 shows the consistent results from this prospective study. Please clarify.” Specifically, Figure 2 showes the proportion of SEs/AEs after first and second dose, while in line 291-296 we discuss about the increased risk for developing SEs/AEs in women compared to men.
6) We thank Reviewer#4 for his comment. In Figure 1 we reported that 74 patients did not receive the second dose due to several, mostly cancer-related, events, including cancer-related deaths. In “results - patients’ characteristics” section, line 111-113, we examined the number of events: the deaths occurred after the first dose were 4 and all were cancer-related. We discuss this topic in the discussion section, as suggested by the Reviewer.
Round 2
Reviewer 2 Report
after making corrections, the manuscript can be qualified for publication without corrections